# Alterations in Plasma Lipid Profile before and after Surgical Removal of Soft Tissue Sarcoma

**DOI:** 10.3390/metabo14050250

**Published:** 2024-04-25

**Authors:** Jae-Hwa Lee, Mi-Ri Gwon, Jeung-Il Kim, Seung-young Hwang, Sook-Jin Seong, Young-Ran Yoon, Myungsoo Kim, Hyojeong Kim

**Affiliations:** 1Department of Molecular Medicine, School of Medicine, Kyungpook National University, Daegu 41944, Republic of Korea; leejh0202@knu.ac.kr (J.-H.L.); miri.gwon@knu.ac.kr (M.-R.G.); wintersj@knu.ac.kr (S.-J.S.); yry@knu.ac.kr (Y.-R.Y.); 2BK21 FOUR KNU Convergence Educational Program of Biomedical Sciences for Creative Future Talents, School of Medicine, Kyungpook National University, Daegu 41944, Republic of Korea; 3Clinical Omics Institute, School of Medicine, Kyungpook National University, Daegu 41405, Republic of Korea; 4Department of Orthopaedic Surgery and Biomedical Research Institute, School of Medicine, Pusan National University, Busan 49241, Republic of Korea; kiimji@pusan.ac.kr; 5Pharmacokinetics Laboratory, Clinical Trial Center, Pusan National University Hospital, Busan 49241, Republic of Korea; morajara@naver.com; 6Department of Clinical Pharmacology and Therapeutics, Kyungpook National University Hospital, Daegu 41944, Republic of Korea; 7Department of Neurosurgery, School of Medicine, Kyungpook National University, Daegu 41944, Republic of Korea; aldtn@knu.ac.kr; 8Department of Internal Medicine, Division of Hemato-Oncology, Maryknoll Hospital, Busan 48972, Republic of Korea

**Keywords:** soft tissue sarcoma, metabolomics, lipid, recurrence, surgery

## Abstract

Soft tissue sarcoma (STS) is a relatively rare malignancy, accounting for about 1% of all adult cancers. It is known to have more than 70 subtypes. Its rarity, coupled with its various subtypes, makes early diagnosis challenging. The current standard treatment for STS is surgical removal. To identify the prognosis and pathophysiology of STS, we conducted untargeted metabolic profiling on pre-operative and post-operative plasma samples from 24 STS patients who underwent surgical tumor removal. Profiling was conducted using ultra-high-performance liquid chromatography–quadrupole time-of-flight/mass spectrometry. Thirty-nine putative metabolites, including phospholipids and acyl-carnitines were identified, indicating changes in lipid metabolism. Phospholipids exhibited an increase in the post-operative samples, while acyl-carnitines showed a decrease. Notably, the levels of pre-operative lysophosphatidylcholine (LPC) O-18:0 and LPC O-16:2 were significantly lower in patients who experienced recurrence after surgery compared to those who did not. Metabolic profiling may identify aggressive tumors that are susceptible to lipid synthase inhibitors. We believe that these findings could contribute to the elucidation of the pathophysiology of STS and the development of further metabolic studies in this rare malignancy.

## 1. Introduction

Soft tissue sarcoma (STS) is a heterogeneous disease entity with approximately 70 subtypes, despite its prevalence being only 1% among adult malignancies [1]. STS originates from mesenchymal cells found in connective tissues such as muscles, blood vessels, neurons, cartilage, and adipose tissue. The scarcity of cases and the absence of large-scale randomized controlled trials pose significant challenges in diagnosing and treating these rare malignancies.

Surgery is the standard curative treatment for localized STS, either alone or in combination with radiation therapy before and after the surgery [2]. The 5-year survival rate of STS is around 50%, but in cases of metastases, it rapidly decreases to around 10% [2,3]. The prevalence of STS in young adults is relatively high compared to epithelial cancers. Therefore, it is necessary to study the prognosis after surgery and understand the mechanisms of development and recurrence of STS to improve survival rates.

The Cancer Genome Atlas (TCGA) program and its accompanying studies have provided a wealth of valuable information about cancer. It is now believed that the malignancy of human bodies is primarily a result of oncogenic mutations, and there have been successful drugs targeting these mutations. However, STS cases exhibit a lower oncogenic mutational burden compared to other epithelial malignancies in the solid tumor categories [4,5]. The lower mutational burden in STS is an unmet need in this era of TCGA and targeted therapies. Therefore, it is necessary to go beyond genomic alterations and explore the metabolic profiles to better understand the pathophysiology especially in STS. The tumor microenvironment (TME) has gained importance in the treatment of malignancies. The metabolic rewiring of tumors can be considered a part of the TME and a result of the interaction between tumor cells and the TME.

Metabolomics is the study of metabolites found in bio-fluids, tissues, and organisms. As metabolites are the byproducts of cellular processes, metabolomics provides a snapshot of the physiological state of an organism. Indeed, metabolites can be promising biomarkers associated with various diseases, especially cancers [6,7]. In particular, untargeted metabolomics is advantageous as it allows for an unbiased analysis of metabolomes derived from various metabolic pathways. A previous study has reported physiological alterations in the serum of colorectal cancer (CRC) patients before and after surgery through untargeted and targeted metabolomics [8]. This type of research design, which compares pre-operative and post-operative profiling, can shed light on the process of tumorigenesis. Furthermore, metabolomics can provide insights into patient prognosis by evaluating metabolic profiles. A study has shown a prognostic nomogram that included metabolic profiles in gastric cancer patients [9].

However, only a limited number of studies have investigated the metabolites in pre-operative and post-operative samples, especially in rare malignancies such as STS. In this study, we examined the metabolome in plasma samples of STS patients before and after surgery using untargeted metabolomics based on ultra-high-performance liquid chromatography–quadrupole time-of-flight/mass spectrometry (UHPLC-QTOF/MS). We aimed to gain valuable insights that could contribute to the early diagnosis of the disease or relapse, as well as to shed light on the pathophysiology of STS.

## 2. Materials and Methods

### 2.1. Reagents and Chemicals

High-performance liquid chromatography (HPLC) grade was used for analysis. Ultrapure distilled water and acetonitrile were purchased from J.T. Baker^®^ (Avantor Performance Materials, LLC., Radnor, PA, USA). Methanol was purchased from Merck (Darmstadt, Germany). Formic acid (LC-MS grade, >98.0%) was purchased from Tokyo Chemical Industry Co., Ltd. (Tokyo, Japan). Hexakis (2,2-difluoroethoxy) phosphazene for lock-mass was purchased from Apollo Scientific Ltd. (Bredbury, UK). Sodium formate solution (10 mM sodium hydroxide with 0.2% formic acid in isopropanol/water, 1:1 *v*/*v*) for internal calibration was purchased from Honeywell Fluka™ (Charlotte, NC, USA).

### 2.2. Collection and Preparation of Samples

This was a prospective study that enrolled 36 patients who underwent surgical resection of STS from November 2018 to September 2021 at Pusan National University Hospital. Blood samples were collected from subjects before and after the operation. The collected blood samples were centrifuged for 10 min each and then immediately stored frozen at −70 °C. Patients who did not meet the inclusion criteria were excluded, including cases where benign tumors were diagnosed based on histopathological findings (*n* = 3), cases where carcinoma was confirmed pathologically with surgical specimens (*n* = 1), cases where post-operative samples were not collected (*n* = 6), and cases where no tumor was visible during surgery (*n* = 2) (Figure 1).

We grouped STSs based on chemosensitivity using the UK guidelines and reported the data [10,11]. Chemosensitivity refers to the responsiveness to chemotherapies. Although the chemotherapy differs according to the subtype, the main drugs commonly used are doxorubicin and ifosfamide. These guidelines categorized STS into five groups based on chemosensitivity: (1) chemotherapy integral to management, (2) chemosensitive, (3) moderately chemosensitive, (4) relatively chemo-insensitive, and (5) chemo-insensitive.

This prospective study was approved by the Institutional Review Board (IRB) of Pusan National University Hospital, with the requirement for written consent (IRB 1805-028-067). The study was performed in accordance with relevant guidelines and regulations. Plasma samples were prepared in 50 μL aliquots, and 100 μL of cold acetonitrile was added for protein precipitation. The samples were mixed thoroughly and centrifuged at 16,100× *g* for 15 min at 4 °C. After drying the supernatant of 100 μL using a vacuum concentrator for 2.2 h, 200 μL of 50% acetonitrile was added to the residuals.

### 2.3. Metabolomics Analysis

The metabolomics analysis was performed using a Thermo Scientific Dionex UltiMate 3000 UHPLC (Dionex Softron GmbH, Germering, Germany) with a Waters ACQUITY UPLC^®^ BEH C18 column (100 mm × 2.10 mm, 1.7 μm, 130 Å; Waters, Milford, MA, USA) coupled to compact QTOF (Bruker Daltonics GmbH & Co. KG, Bremen, Germany). Separation was conducted at a flow rate of 300 μL/min using a mobile phase consisting of 0.1% formic acid in water (A) and acetonitrile (B). The gradient used was as follows: 1% B, 0.0–1.0 min; 1–65%, 0.5–3.0 min; 65–90%, 3.0–7.0 min; 90%, 7.0–35.0 min; 90–100%, 35.0–35.5 min; 100%, 35.0–41.5 min. The gradient then returned to the initial concentration (1% B) for 2 min before the next sample. The auto-sampler and column temperature were maintained at 4 °C and 40 °C, respectively. The injection volume was 1.5 μL.

The mass spectrometer was operated in the positive ionization mode for mass measurement, using the following parameters: mass scan range, full scan 50–1000 mass-to-charge ratio (*m*/*z*); nebulizer gas pressure, 0.8 bar; capillary voltage, +4500 V; end plate offset, −500 V; dry gas flow rate, 10.0 L/min; dry gas temperature, 200 °C.

### 2.4. Putative Identification of Metabolites

The tandem mass (MS/MS) spectrum was compared to the libraries of MetaboScape 5.0 (Bruker Daltonics GmbH & Co. KG, Bremen, Germany), such as the Human Metabolome Database (HMDB) Metabolite Library, MetaboBASE Personal Library, and MS-DIAL LipidBlast (version 68). The annotation parameters were as follows: mass tolerance, 2.0–5.0 mDa; mSigma, 50–100. The mSigma is a measure of the goodness of fit between the measured and theoretical isotopic patterns.

### 2.5. Statistical Analysis

ProfileAnalysis 2.1 (Bruker Daltonics, Billerica, MA, USA) was used to construct the feature table. The raw data were preprocessed by performing quantile normalization, log transformation, and pareto scaling. SIMCA 17.0.2 (Sartorius Stedim Data Analytics AB, Umeå, Sweden) was employed for multivariate statistical analysis, such as principal component analysis (PCA) and orthogonal projections to latent structures–discriminant analysis (OPLS-DA). To verify the OPLS-DA results, a permutation test with 100 iterations was implemented. As the variable importance in projection (VIP) value represents the contribution of each feature, metabolites with high VIP values are more relevant for group separation [12]. A VIP value of 1.0 or higher was considered significant. A paired *t*-test was conducted to evaluate the differences in metabolites between the before and after surgery groups using SPSS Statistics 26.0 (IBM Corp., Armonk, NY, USA).

To identify the metabolic signature contributing to group discrimination and evaluate the predictive performance of potential biomarkers in distinguishing recurrence of STSs, the univariate receiver operator characteristic (ROC) curve analysis was performed. For the ROC curve, the area under the curve (AUC) was calculated to assess the accuracy of the metabolites. A general guide was used to estimate the accuracy based on AUC values: 0.5–0.6, fail; 0.6–0.7, poor; 0.7–0.8, fair; 0.8–0.9, good; and 0.9–1.0, excellent [13]. MetaboAnalyst version 5.0 (https://www.metaboanalyst.ca, accessed on 4 December 2023) was used to perform the ROC curve analyses.

## 3. Results

### 3.1. Patient Characteristics

We conducted a study on 24 patients who had 10 different pathological subtypes of STS (Table 1). The median age of the patients, consisting of 11 females and 13 males, was 61 years (ranging from 42 to 76 years) at the time of diagnosis. The majority of the primary locations of the tumors were in the extremities, except for one case of dedifferentiated liposarcoma in the retroperitoneum (Table 1). Appendix A provides more detailed individual information. Among these 24 patients with STS, leiomyosarcoma (six patients) was the most common subtype, followed by myofibrosarcoma (five patients).

Thirteen out of twenty-four patients had relatively chemo-insensitive STSs, which included myxofibrosarcoma, dedifferentiated liposarcoma, well-differentiated liposarcoma, undifferentiated pleomorphic sarcoma, and malignant peripheral nerve sheath tumor (MPNST). Moderately chemosensitive STS cases included leiomyosarcoma, angiosarcoma, and pleomorphic liposarcoma. Only 2 out of 24 patients with myxoid liposarcoma had chemosensitive STS. Pleomorphic leiomyosarcoma could not be evaluated due to very recently separate categorization from leiomyosaroma and limited data on chemosensitivity. The median relapse-free survival (RFS) for these patients was 4 years.

### 3.2. Metabolite Profiles of Sarcoma Patients

The pre-operative and post-operative plasma samples were analyzed using untargeted metabolomics profiling. The quality control samples were clustered together in the PCA score plot, and there was a slight separation observed between pre-operative and post-operative samples, although it was not very clear (Figure 2). The OPLS-DA score plot clearly differentiated between pre-operative and post-operative STS samples, with an R2Y value of 0.971 and a Q2 value of 0.519 (Figure 3A). In the permutation test to validate the OPLS-DA model, the y-intercept of the R2 and Q2 regression lines were 0.963 and −0.232, respectively (Figure 3B).

### 3.3. Metabolite Profiles of Sarcoma Patients

Based on the VIP value obtained from the OPLS-DA model, a total of 39 metabolites were screened (Table 2). Using either the paired *t*-test or Wilcoxon Rank–Sum test, it was found that 34 metabolites exhibited statistical significance. Among these, nine metabolites showed downregulation, while the rest exhibited upregulation in postoperative STS patients. The trends of these metabolites are illustrated on a heatmap (Figure 4).

Most of the putatively identified metabolites were lipids, specifically glycerophospholipids such as phosphatidylcholine (PC), lysophosphatidylcholine (LPC), phosphatidylethanolamine (PE), lysophosphatidylethanolamine (LPE), and lysophosphatidylserine (LPS), as well as fatty acids (FAs) and their derivatives. These metabolites are involved in FA and glycerophospholipid metabolism.

### 3.4. Analysis of Receiver Operating Characteristics for Potential Biomarkers

After the operation, 11 out of 24 patients experienced a recurrence. To identify potential biomarkers that contribute to recurrence in STS patients, a univariate analysis was conducted. The pre-operative and post-operative plasma samples were classified into the recurrence and non-recurrence subgroups. Subsequently, metabolites showing significant differences between the recurrence and non-recurrence subgroups were selected. The levels of LPC O-18:0 and LPC O-16:2 in pre-operative plasma were significantly lower in patients who experienced the recurrence after the operation compared to those who did not, with *p*-values of 0.044 and 0.014, respectively (Figure 5A,B).

To further assess the predictive potential of LPC O-18:0 and LPC O-16:0 for the recurrence of STSs, we conducted a univariate ROC analysis, which allowed us to obtain information about the sensitivity and specificity of these potential biomarkers. The AUC values for LPC O-18:0 and LPC O-16:2 were 0.748 and 0.797, respectively. The AUC values, with a 95% confidence interval, are depicted in Figure 5C,D. The corresponding sensitivity and specificity values are provided in Table 3.

## 4. Discussion

In this study, we utilized UHPLC-QTOF/MS to perform untargeted metabolomics in the plasma of STS patients and found differences in the metabolic profiles between pre-operative and post-operative STS patients. This finding is important for understanding the underlying pathology of STS, as valuable insights can be obtained from analyzing plasma metabolites. Particularly, the results highlight metabolic alterations that reflect the genome, transcriptome, and proteome. Additionally, comparing the preoperative plasma metabolites with those after the operation allows for the evaluation of the standard treatment approach for STS, which involves surgical resection of the tumor mass. This comparison also provides insight into predicting the prognosis of STS after surgery.

The most significantly altered metabolic profiles in the post-operative plasma of STS patients were observed in phospholipids, particularly PC and LPC, which exhibited a considerable increase. PC is the most abundant phospholipid in mammalian cells, accounting for 40–50% of total cellular phospholipids [14]. It plays a crucial role in biological membranes and is involved in cell division, growth, and the synthesis of lipoproteins, which are responsible for lipid transport. Phospholipids are particularly crucial for the formation and release of extracellular vesicles (EVs), facilitating bidirectional cell-to-cell communication within TME [15]. The significance of EVs in tumor angiogenesis has been widely recognized [16,17]. Microvesicles from sarcoma patients enhance tumor vascularization by facilitating receptor translocation, increasing intracellular calcium levels, mitochondrial activity, and adenosine triphosphate production [16]. Additionally, osteosarcoma-derived EVs promote angiogenesis by transferring pro-angiogenic proteins and miRNAs to the epithelial cells [17]. However, it is worth noting that despite the potential involvement of phospholipids in EV formation, lower levels of phospholipids were observed in the pre-operative stage compared to the post-operative stage. Since our study did not include ultra-centrifugation, it was uncertain whether the observed metabolites reflect those associated with tumor-derived EVs. Further investigation is needed to better understand the dynamics of lipid metabolism in STS-derived EVs and its potential implications for metastasis.

In our study, the putatively identified metabolites were found to be involved in FA and glycerophospholipid metabolism. The altered metabolic pathways suggest a potential reprogramming of lipid metabolism in STS. Malignant tumors have a significant requirement for lipids in cell membranes and energy metabolism to support critical processes associated with cell growth, proliferation, invasion, and angiogenesis. This alteration in lipid metabolism is regarded as a hallmark of aggressive tumors [18,19]. Tumors also have a high ability to adapt to their environment, enabling them to continue growing and to survive even in unfavorable conditions, such as hypoxic conditions. Hypoxia, commonly observed in tumors such as sarcomas, leads to the upregulation of hypoxia-inducible factors (HIFs). This hypoxia-induced stabilization of HIFs drives the transcription of over 150 genes, including carbonic anhydrase 9, glucose transporter 1, and vascular endothelial growth factor [20,21], which play key roles in angiogenesis, metabolic reprogramming, invasion, metastasis, and resistance to radiation therapy and chemotherapy [22]. Throughout these processes, FAs and glycerophospholipids emerge as indispensable components.

The reprogramming of lipid metabolism can also impact the lipid composition in the bloodstream. Numerous studies have been conducted on alterations in lipid metabolism in different types of solid tumors. For example, in the plasma of endometrial cancer patients, several PCs have been found to be significantly decreased compared to controls [23], and a decrease in plasma phospholipids has also been observed in lung cancer patients through untargeted metabolomics [24]. In CRC patients, a reduced plasma level of LPC has been reported, suggesting its potential as a biomarker for CRC [25]. The decreased levels of phospholipids in the plasma of patients with tumors may indicate that lipids are transferred from the bloodstream to the tumor due to the high lipid consumption by tumor cells [26]. In addition, a study investigating gene alterations related to metabolic pathways across 32 different cancer types has suggested that the most pronounced genetic variations in sarcomas are associated with lipid metabolism [27]. The same study has also indicated that cancers with a higher frequency of genetic alterations in metabolic genes show shorter survival rates compared to those with fewer alterations. The FA metabolism-related genes in STS were abnormally expressed [28]. In our study, we found that the levels of PC and LPC in pre-operative STS patients were lower than those in post-operative patients. This suggests that STS cells present in the body consume excess amounts of lipids for their survival. Moreover, the increase in phospholipid levels after tumor removal can be attributed to the disappearance of the primary consumer [29].

The high levels of plasma acyl-carnitines observed in pre-operative STS patients are presumed to be associated with an increased energy supply caused by metabolic changes within tumor cells. Acyl-carnitines are conjugations of FAs with carnitine and serve as carriers transporting FAs to the mitochondrial matrix. This transportation facilitates fatty acid beta-oxidation (FAO) within cells, playing a crucial role in energy metabolism to sustain cell activity [30]. Acyl-carnitines have been implicated in various disease states, including insulin resistance [31], obesity [32], breast cancer [33], hepatocellular carcinoma [34], and nonalcoholic fatty liver disease (NAFLD) [35,36]. In NAFLD patients, the serum levels of total acyl-carnitine increased gradually according to the progression of fibrosis, with even higher levels in hepatocellular carcinoma patients [36]. The high levels of acyl-carnitines in pre-operative samples in this study could indicate lipid metabolism reprogramming as well. These altered profiles suggest a potential shift in the utilization of lipid resources and may also reflect the interplay between the tumor cells and TME in energy metabolism.

FAO has been shown to be abnormally active in various tumors, which are closely related to the proliferation, metastasis, and resistance to chemotherapy of tumor cells [37]. In lipid metabolism, acyl-CoA synthetases (ACSL) convert long-chain FAs into fatty acyl-CoA esters. Subsequently, these esterified forms are further converted into acyl-carnitine by carnitine palmitoyltransferase 1 (CPT1). A recent study indicated that transforming growth factor beta 1 treatment induces ACSL3 upregulation, promoting lipid metabolic reprogramming in CRC cells through the activation of the FAO pathway [38]. A British research team reported that ACSL3 and ACSL4 were highly expressed in STS cells. They found that the expression levels differed according to the subtype of STS, with the expression increasing in the order of liposarcoma, fibrosarcoma, leiomyosaroma, and rhabdomyosarcoma. Although the study was conducted with cell lines, we believe this difference could be associated with the chemosensitivity in vivo [39]. MPNST is known as a relatively chemo-insensitive STS with a poor prognosis. An American research team studied FA synthase as a metabolic target in this STS and observed that MPNST cells accumulated lipid droplets, while the inhibition of FAO decreased oxygen consumption and reduced MPNST viability [40]. Among the three subtypes of CPT1, elevated levels of CPT1C mRNA have been reported in specific cancer types, particularly in STSs, with Ewing’s sarcoma and bone sarcoma following in rank [41]. This suggests a shift in lipid metabolic reprogramming towards the FAO pathway to meet energy demands during invasion of these malignancies.

In the ROC analysis results, LPC O-18:0 and LPC O-16:2 emerged as significant metabolites in distinguishing between recurrent and non-recurrent patients, demonstrating their potential as prognostic biomarkers for assessing the risk of recurrence post-operation. A Korean study has revealed that tumor metastasis to lymph nodes requires a metabolic shift towards FA oxidation. This suggests a correlation between lipid metabolism and more aggressive tumors [42]. Aggressive STS might have consumed more LPCs. The results not only provide treatment strategies tailored to individual patients, potentially enhancing clinical outcomes, but also underscore the significance of targeting lipid metabolism in the strategy of STS treatment.

This study has some limitations. The most disappointing factors are the small number of subjects due to the rarity of STS and the failure to validate the results in healthy individuals. Validation in healthy individuals would have ensured the reliability of the results and allowed for a more accurate assessment of the surgical effect. However, we believe that this study is valuable considering the rarity of the disease. The second limitation is the timing of collecting post-operative samples. The intervals between the operation and post-operative sample collection ranged from 5 days to 45 days (Appendix A). The alterations of metabolic profiles might be attributable to the operation itself. It has been reported that lipid metabolism can be affected by the circadian rhythm, and the feeding/fasting cycle affects the circadian system [43,44]. However, only one patient had a sample acquired 5 days after the operation, and most samples were collected at least 7 days after the operation. We should have controlled the interval more homogenously, although 18 out of 24 patients offered their post-operative samples between 7 and 14 days after the surgeries.

We conducted survival analyses as well. The Kaplan–Meier curve showed distinct RFS lines based on the chemosensitivity and pre-operative chemotherapy. However, the number of patients was too small to yield statistically significant results. The observed *p*-values were around 0.15. In addition, we conducted *t*-tests and univariate ROC curve analyses to identify metabolic markers using the levels of 34 putative metabolites from pre-operative samples. Remarkably, 13Z-Docosenamide showed an AUC value exceeding 0.7 (0.707), indicating its potential significance as a metabolic marker. While the *t*-test did not indicate statistical significance, these findings suggest the need for further exploration and validation of its discriminatory power.

## 5. Conclusions

In conclusion, we characterize endogenous metabolite alterations in the pre-operative and post-operative plasma of STS patients. We observed a significant increase in the plasma levels of PC and LPC after the removal of the STS mass. This suggests a potential transfer of phospholipids from the blood to the tumor tissue in the presence of a tumor The levels of phospholipids increased after the tumor was removed because the primary consumer vanished.

Our findings have potential to enhance the pathophysiological understanding of STS. Furthermore, a noteworthy discovery in this study was the identification of LPC O-18:0 and LPC O-16:2 as potential biomarkers for predicting the prognosis of STS. This result not only suggests a poor prognosis but also indicates susceptibility of aggressive STSs to FA synthesis inhibitors such as TVB-2640 [45]. However, further investigation is necessary to assess the clinical significance of LPC levels in larger cohorts. We recommend creating cohorts with groups based on the chemosensitivity of STS, taking into account the rarity of this disease and the results of previous studies, including this one.

## Figures and Tables

**Figure 1 metabolites-14-00250-f001:**
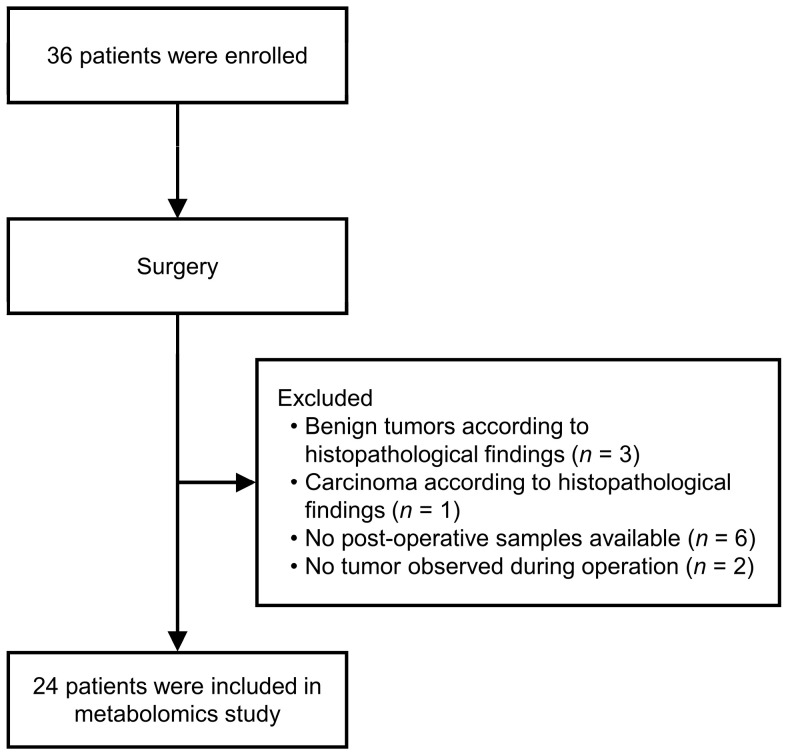
Flowchart of the subject selection process.

**Figure 2 metabolites-14-00250-f002:**
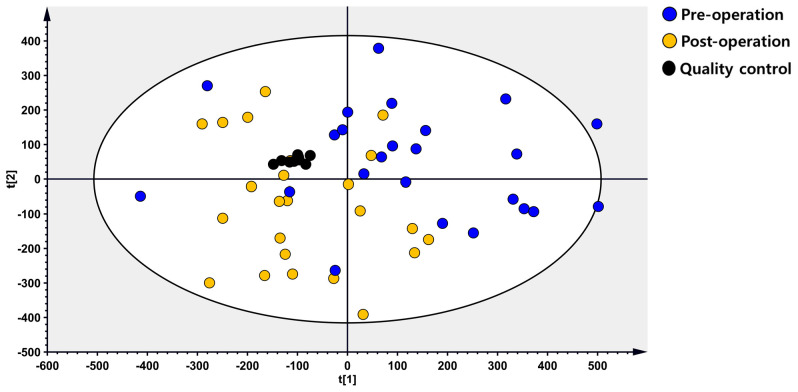
The PCA score plot for pre-operative and post-operative STS samples.

**Figure 3 metabolites-14-00250-f003:**
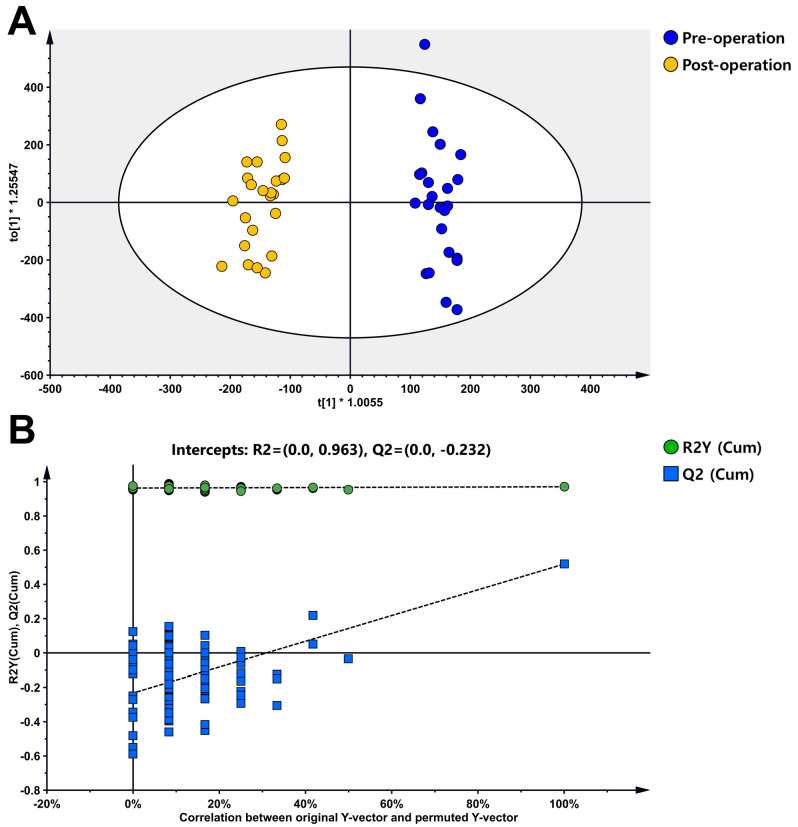
The score and permutation test plot of the OPLS-DA. (**A**) OPLS-DA score plot. R2Y = 0.971, Q2 = 0.519. (**B**) Permutation test plot of the OPLS-DA model (*n* = 100).

**Figure 4 metabolites-14-00250-f004:**
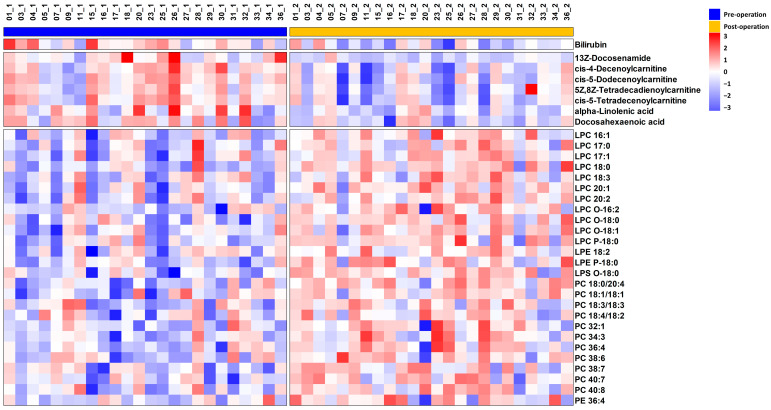
The heatmap of 34 metabolites exhibiting significant alterations between pre-operative and post-operative STS patients.

**Figure 5 metabolites-14-00250-f005:**
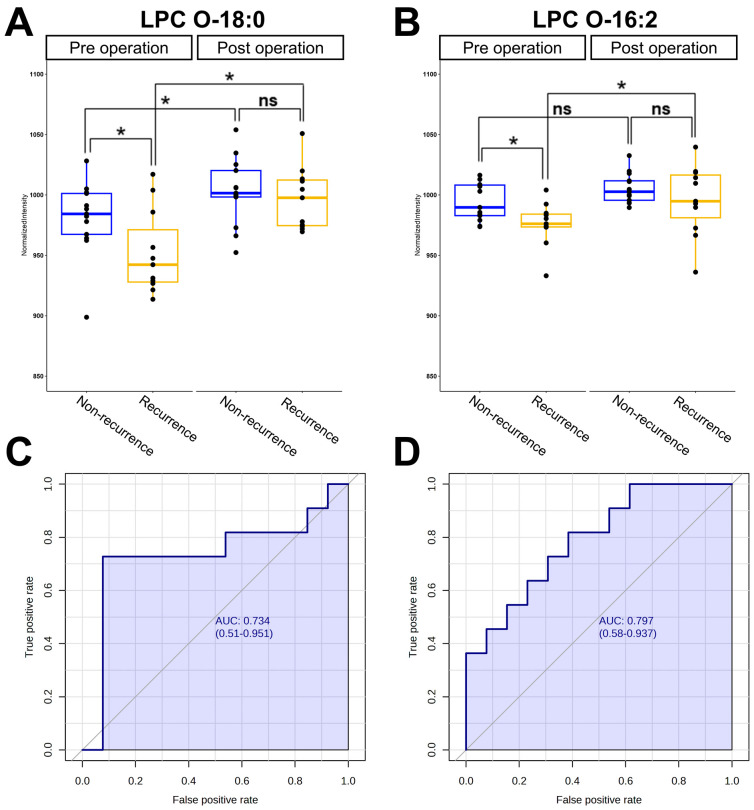
Univariate analysis-based predictive potential of LPC O-18:0 and LPC O-16:2 in distinguishing between recurrence and non-recurrence STS patients at the pre-operative stage. (**A**,**B**) Box plots representing the levels of LPC O-18:0 and LPC O-16:2. (**C**,**D**) Univariate ROC analysis for LPC O-18:0 and LPC O-16:2, presenting the AUC and 95% confidence interval. *, *p* < 0.05; ns, not significant.

**Table 1 metabolites-14-00250-t001:** Pathological characteristics of STS patients.

Pathology	Chemosensitivity	Anatomical Location of Primary Lesion	Patients (*n*)
Angiosarcoma	Moderately chemosensitive	Hip	1
Dedifferentiated liposarcoma	Relatively chemo-insensitive	Calf, thigh, retroperitoneum	3
Leiomyosarcoma	Moderately chemosensitive	Upper arm, hip, thigh (4)	6
MPNST	Relatively chemo-insensitive	Shoulder	1
Myxofibrosarcoma	Relatively chemo-insensitive	Upper arm, forearm, thigh (3)	5
Myxoid liposarcoma	Chemosensitive	Thigh	2
Pleomorphic leiomyosarcoma	NE	Thigh	1
Pleomorphic liposarcoma	Moderately chemo-sensitive	Thigh	1
Undifferentiated pleomorphic sarcoma	Relatively chemo-insensitive	Calf, hip, thigh	3
Well differentiated liposarcoma	Relatively chemo-insensitive	Hip	1

Numbers in parentheses represent the number of patients with missing data. MPNST, malignant peripheral nerve sheath tumor; NE, not evaluable.

**Table 2 metabolites-14-00250-t002:** List of differential metabolites in plasma samples between pre-operative and post-operative STS patients.

Metabolites	VIP	RT (min)	*m*/*z*	Formula	Trend ^a^	*p*-Value
Porphyrin Metabolism; Bile Secretion						
Bilirubin	2.62	5.28	585.2713	C_33_H_36_N_4_O_6_	↓	0.003
Fatty Acid Metabolism						
N-Palmitoyl threonine	1.00	5.98	358.2932	C_20_H_39_NO_4_	↓	0.245
13Z-Docosenamide	1.34	12.96	338.3430	C_22_H_43_NO	↓	0.004
Nervonamide	1.20	15.89	366.3738	C_24_H_47_NO	↓	0.072
cis-4-Decenoylcarnitine	2.63	5.30	314.2328	C_17_H_31_NO_4_	↓	0.002
cis-5-Dodecenoylcarnitine	3.04	5.64	342.2636	C_19_H_35_NO_4_	↓	0.001
5Z,8Z-Tetradecadienoylcarnitine	2.63	5.79	368.2790	C_21_H_37_NO_4_	↓	0.022
cis-5-Tetradecenoylcarnitine	3.19	6.04	370.2948	C_21_H_39_NO_4_	↓	0.001
alpha-Linolenic acid	1.38	9.13	279.2312	C_18_H_30_O_2_	↓	0.009
Docosahexaenoic acid	2.39	9.38	329.2480	C_22_H_32_O_2_	↓	0.014
Glycerophospholipid Metabolism						
LPC 16:1	1.48	6.84	494.3263	C_24_H_48_NO_7_P	↑	0.001
LPC 17:0	1.16	8.14	510.3557	C_25_H_52_NO_7_P	↑	0.002
LPC 17:1	1.30	7.31	508.3422	C_25_H_50_NO_7_P	↑	0.003
LPC 18:0	1.20	8.56	524.3714	C_26_H_54_NO_7_P	↑	0.000
LPC 18:3	1.53	6.65	518.3248	C_26_H_48_NO_7_P	↑	0.026
LPC 20:1	1.78	8.98	550.3888	C_28_H_56_NO_7_P	↑	0.000
LPC 20:2	1.37	8.09	548.3735	C_28_H_54_NO_7_P	↑	0.005
LPC O-16:2	1.28	30.95	478.3303	C_24_H_48_NO_6_P	↑	0.007
LPC O-18:0	1.65	9.19	510.3930	C_26_H_56_NO_6_P	↑	0.000
LPC O-18:1	1.62	9.14	508.3753	C_26_H_54_NO_6_P	↑	0.001
LPC P-18:0	1.18	8.13	508.3740	C_26_H_54_NO_6_P	↑	0.002
LPE 18:2	1.01	7.10	478.2935	C_23_H_44_NO_7_P	↑	0.011
LPE 22:5	1.19	7.25	528.3093	C_27_H_46_NO_7_P	↑	0.068
LPE P-18:0	1.79	9.12	466.3305	C_23_H_48_NO_6_P	↑	0.000
LPS O-18:0	1.68	6.80	512.3363	C_24_H_50_NO_8_P	↑	0.001
PC 16:0/20:5	1.24	17.81	780.5541	C_44_H_78_NO_8_P	↑	0.500
PC 18:0/20:4	1.86	25.36	810.6000	C_46_H_84_NO_8_P	↑	0.000
PC 18:1/18:1	1.02	31.71	786.5999	C_44_H_84_NO_8_P	↑	0.018
PC 18:2/18:3	1.21	15.43	780.5511	C_44_H_78_NO_8_P	↑	0.069
PC 18:3/18:3	1.71	15.31	778.5419	C_44_H_76_NO_8_P	↑	0.001
PC 18:4/18:2	1.11	18.75	778.5348	C_44_H_76_NO_8_P	↑	0.039
PC 32:1	1.48	22.76	732.5541	C_40_H_78_NO_8_P	↑	0.001
PC 34:3	1.87	19.57	756.5552	C_42_H_78_NO_8_P	↑	0.003
PC 36:4	1.17	31.18	782.5684	C_44_H_80_NO_8_P	↑	0.007
PC 38:6	1.14	25.01	806.5683	C_46_H_80_NO_8_P	↑	0.003
PC 38:7	1.17	15.53	804.5562	C_46_H_78_NO_8_P	↑	0.000
PC 40:7	1.11	19.64	832.5880	C_48_H_82_NO_8_P	↑	0.000
PC 40:8	1.51	16.12	830.5694	C_48_H_80_NO_8_P	↑	0.001
PE 36:4	1.45	22.07	740.5210	C_41_H_74_NO_8_P	↑	0.014

^a^ The trend refers to an increase or decrease in metabolite levels in the plasma of post-operation compared to pre-operation. The upper and lower arrows indicate an increase or decrease after surgery, respectively. A *p*-value below 0.05 was considered to indicate a significant difference. VIP, variable importance in projection; RT, retention time; *m*/*z*, mass-to-charge ratio; LPC, lysophosphatidylcholine; LPE, lysophosphatidylethanolamine; LPS, lysophosphatidylserine; PC, phosphatidylcholine; PE, phosphatidylethanolamine.

**Table 3 metabolites-14-00250-t003:** The AUC values obtained from univariate ROC curve analyses of the LPC O-18:0 and LPC O-16:2 (confidence intervals are shown in brackets) along with their sensitivity and specificity.

Metabolites	AUC Value	Sensitivity	Specificity
LPC O-18:0	0.734 (0.510–0.951)	0.727	0.923
LPC O-16:2	0.797 (0.580–0.937)	0.727	0.615

The 95% confidence interval was calculated using 500 bootstrappings and is provided in parentheses. LPC, lysophosphatidylcholine; AUC, area under the curve.

## Data Availability

The data presented in this study are available on request from the corresponding author. The data are not publicly available due to restriction of privacy.

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
