# Peer review of "Alterations in Plasma Lipid Profile before and after Surgical Removal of Soft Tissue Sarcoma"

_metabolites, 2024, doi:10.3390/metabo14050250_

Round 1

Reviewer 1 Report

Comments and Suggestions for Authors

The paper by Lee et al revealed distinct lipid profile alterations in STS patients after surgical removal of masses.  potentially contribute to the elucidation of the pathophysiology of STS 

 Although the  study utilizes several patients  n=24, the PC lipids particularly represented in post-operative STS  no confirmative data are presented

the  discussion should  include  the role of these lipids which could be also to generate vesicles   to transport miRNA or proteins in another body site

favoring metastasis add doi: 10.1038/s41419-021-04069-w.

Lane 53 STS are mesenchymal tumors

Comments on the Quality of English Language

Dear Editor, the paper by Lee et al revealed distinct lipid profile alterations in STS patients after surgical removal of masses.  potentially contribute to the elucidation of the pathophysiology of STS 

 Although the  study utilizes several patients  n=24, the PC lipids particularly represented in post-operative STS  no confirmative data are presented

the  discussion should  include  the role of these lipids which could be also to generate vesicles   to transport miRNA or proteins in another body site

favoring metastasis add doi: 10.1038/s41419-021-04069-w.

Reviewer 2 Report

Comments and Suggestions for Authors

This article addresses a rare and therefore less studied type of cancer. The authors observed significant differences in the plasma metabolome of patients before and after surgery, in relation to lipid content. The main criticisms I would make would be regarding the lack of control samples from healthy subjects and the differences in sample collection time after surgery. It is very difficult to analyze the influence of wound healing in the post- surgery results if time is so variable. However, the authors themselves recognize these limitations in their study. Despite this, I consider that the article has merit and deserves to be published, given the fact that there are few studies on sarcomas. I suggest that the authors include a table (it could be as supplementary data) with information about the post-surgery collection time for each individual, as well as chemotherapy treatment and time when recurrence was detected.

Minor points:

1. There are some repeated sentences such as the ones in lines 28-30, 49-51 and 109-111. The text could be improved by combining these sentences.

2. Line 100, please inform that the plasma was frozen.

3. Lines 110-112. Chemosensitivity to which drugs? Please specify.

Reviewer 3 Report

Comments and Suggestions for Authors

Comments on the Quality of English Language

No, thank you!

Round 2

Reviewer 3 Report

Comments and Suggestions for Authors

No further comments on the current version.

Comments on the Quality of English Language

No.